# Mixing Performance of a Passive Micro-Mixer with Mixing Units Stacked in Cross Flow Direction

**DOI:** 10.3390/mi12121530

**Published:** 2021-12-09

**Authors:** Makhsuda Juraeva, Dong-Jin Kang

**Affiliations:** School of Mechanical Engineering, Yeungnam University, Gyoungsan 712-749, Korea; mjuraeva@ynu.ac.kr

**Keywords:** passive micro-mixer, mixing unit, cross flow direction, baffle impingement, swirl motion, mixing performance

## Abstract

A new passive micro-mixer with mixing units stacked in the cross flow direction was proposed, and its performance was evaluated numerically. The present micro-mixer consisted of eight mixing units. Each mixing unit had four baffles, and they were arranged alternatively in the cross flow and transverse direction. The mixing units were stacked in four different ways: one step, two step, four step, and eight step stacking. A numerical study was carried out for the Reynolds numbers from 0.5 to 50. The corresponding volume flow rate ranged from 6.33 μL/min to 633 μL/min. The mixing performance was analyzed in terms of the degree of mixing (*DOM*) and relative mixing energy cost (*MEC*). The numerical results showed a noticeable enhancement of the mixing performance compared with other micromixers. The mixing enhancement was achieved by two flow characteristics: baffle wall impingement by a stream of high concentration and swirl motion within the mixing unit. The baffle wall impingement by a stream of high concentration was observed throughout all Reynolds numbers. The swirl motion inside the mixing unit was observed in the cross flow direction, and became significant as the Reynolds number increased to larger than about five. The eight step stacking showed the best performance for Reynolds numbers larger than about two, while the two step stacking was better for Reynolds numbers less than about two.

## 1. Introduction

Mixing at micro scales is an essential design task in the bio- and microfluidic systems such as micro-total analysis system (µ-TAS), lab on a chip, and micro-reactors. As the associated dimensions of a microfluidic system are small, the molecular diffusion is a major mechanism of mixing; the corresponding Reynolds number is also very small and the flow is laminar. Meanwhile, many biochemical processes require rapid and complete mixing, and mixing enhancement is essential in designing these systems [1].

A variety of micro-mixers have been devised to enhance the mixing in a micro-fluidic system [2]. They are named either active or passive micro-mixers, depending on the usage of an external energy source. As active micro-mixers utilize an external energy source to achieve the mixing enhancement, the structure of a micro-mixer becomes more complicated than that of passive micro-mixers. In addition, it is more costly compared with a passive micro-mixer. Examples of external energy sources used for active micro-mixers are acoustic [3], magnetic [4], electric [5], thermal [6], electro [7,8], and pressure fluctuating [9]. On the contrary, passive micro-mixers use micro-mixer geometry to agitate and generate secondary flow. They are expected to promote chaotic advection of fluids leading to the mixing enhancement. Accordingly, the mixing enhancement by a passive micro-mixer could be limited compared with that by a similar, active micro-mixer. However, passive micro-mixers are simpler to fabricate and easier to integrate into microfluidic systems; therefore, they are widely used in microfluidic systems.

Many different types of passive micro-mixers have been designed and investigated to enhance their mixing performance [10]. Some examples include recessed grooves in the channel wall [11], herringbone-type walls [12], baffles [13,14], channel wall twisting [15], junction contraction [16], split and recombine (SAR) [17,18], serpentine microchannel [19,20], converging and diverging microchannel [21], spiral channel [22,23], and periodic geometric features [24].

Among various geometric components, baffles are one of the most widely adopted components in designing a passive micro-mixer. Many researchers have shown that micro-mixers based on baffles perform effectively over a wide range of Reynolds numbers [13,25]. For example, Tsai et al. [25] used radial baffles in a curved microchannel and showed that radial baffles induced vortices in multiple directions. Kang [13] placed rectangular baffles in a cyclic order along the channel wall and enhanced the mixing performance of a T-shaped micromixer. Santana et al. [14] used triangular baffles to obtain a high mixing index for various micro-channel dimensions. Sotowa et al. [26] showed that the indentations and baffles attached to the micromixer wall enhance the mixing by means of secondary flow in deep micro-channel reactors. Chen et al. [27] distributed baffles on both side walls of a micromixer based on the Koch fractal principle. They showed that the baffles distributed on the both side of the microchannel wall performs better than the mixing performance of the baffles distributed on the one side. Chung et al. [28] implemented planar baffles and showed that the mixing performance was enhanced at both of diffusion dominant and convection dominant regime of mixing. Raza et al. [29] improved the mixing performance of a SAR micromixer by embedding baffles just after each SAR unit. The enhancement was shown as achievable over a Reynolds number range from 0.1 to 80.

Recently, several complicated three-dimensional (3D) micro-mixers showed an improved mixing performance compared with those of two-dimensional (2D) micro-mixers of similar size [30,31,32,33]. However, 2D planar micromixers have an advantage of simplicity in fabrication compared with that of complex 3D micromixers. Various design techniques were attempted to promote 3D effects: baffles, channel wall twisting, spiral channel, and split and recombine. For example, Kang [13] showed that a cyclic configuration of baffles generates a rotational flow in the cross section of a micromixer. The same idea was adopted in the present micromixer to promote 3D effects, and four baffles were attached to the four side walls of the micromixer in a cyclic order. The overall layout of a passive micromixer is also an important design concept [33]. For example, Tripathi [23] studied three different layout of a passive micromixer based on spiral microchannel and showed that the mixing performance is quite dependent on the micromixer layout. The eight mixing units were stacked in the cross flow direction; this kind of stacking has not been studied yet. The stacking was designed as four different layouts: one step, two step, four step, and eight step stacking. The combined effects of a cyclic baffle arrangement and mixing unit stacking in the cross flow direction was the main design concept of the present micromixer.

Most of micromixers used in biological and chemical applications usually operate in the range of milliseconds of mixing time, and the corresponding Reynolds number is less than about 100 [34,35,36]; the corresponding volume flow rate is an order of mL/h. In this range, the mixing mechanism can be divided into three regimes: molecular dominance, transition, and advection dominance [10,18,33]. The effects of micromixer layout design as well as a passive device such as baffles are known to be significant in all of the three regimes. Accordingly, a numerical study was carried out for the Reynolds numbers from 0.5 to 50, covering all of the three regimes. The corresponding volume flow rate ranged from 6.33 μL/min to 633 μL/min.

A numerical simulation has several advantages to study the fluid dynamic and mixing features involved in a micro-mixer, including easy visualization of the mixing process and the associated flow characteristics such as streamlines and vortex formation. Accordingly, a numerical approach is widely accepted in studying mixing enhancement mechanism of a micro-mixer. For example, Rhoades et al. [37] used the commercial software COMSOL Multiphysics 5.1 (COMSOL, Inc., Burlington, MA, USA) to study the mixing performance of grooved serpentine microchannels. Volpe et al. [38] studied the flow dynamics of a continuous size-based sorter microfluidic device by using the lattice Boltzmann method (LBM). Kang [13] used the commercial software ANSYS^®^ Fluent 19.2 (ANSYS, Inc., Canonsburg, PS, USA) [39] to simulate the mixing performance of a passive micromixer with baffles and quantitatively evaluated the mixing performance.

In this paper, the commercial software ANSYS^®^ Fluent 19.2 was used to simulate the mixing performance. The mixing performance was estimated in terms of the degree of mixing (*DOM*) and corresponding mixing energy cost (*MEC*). The numerical simulation was carried out for Reynolds numbers ranging from 0.5 to 50, corresponding to volume flow rates ranging from 6.33 μL/min to 633 μL/min.

## 2. Passive Micro-Mixer with Mixing Units Stacked in the Cross Flow Direction

Figure 1 shows a mixing unit with four hexahedron baffles inside. The four baffles were arranged in a cyclic order. The first baffle was attached to the lower wall, and the third baffle was attached to the upper wall. The second and fourth baffles were attached to the front and back walls, respectively. Each baffle was 30 µm thick, and blocked the flow passage by half in each direction. Two consecutive baffles were separated by 60 µm, and each mixing unit was 600 µm long. Figure 2 depicts the five different layouts of a micromixer simulated in this paper. Each micromixer had eight mixing units, and each mixing unit consisted of four baffles. Figure 2a shows the baseline layout of eight mixing units, and Figure 2b–e shows mixing units stacked in the cross stream direction (y direction); the mixing units were overlapped 150 µm in the main stream direction (x direction). The inlets and outlet had a rectangular cross-section of 300 µm × 120 µm. The two inlet branches were 1000 µm long, and the outlet branch was 600 µm long.

For the sake of simplicity, we assumed that the same aqueous solution flows into the two inlets: Inlet 1 and Inlet 2. The properties of the aqueous solution are similar to those used in BioMEMS systems. The kinematic viscosity and mass diffusion coefficient of the aqueous solution are v = 10^−6^ m^2^/s and *D* = 10^−9^ m^2^/s, respectively [18,33]. The corresponding Schmidt (Sc) number is 10^3^; it is defined as the ratio of the kinetic viscosity and the mass diffusion coefficient. The Reynolds number is defined as $Re=\frac{U_{mean}d_{h}}{v}$, where $U_{mean},\;d_{h}\;\mathrm{and}\;v$ indicate the mean velocity at the outlet, the hydraulic diameter of outlet, and the kinematic viscosity of the fluid, respectively.

## 3. Governing Equations and Computation Procedure

Transport phenomena in micromixers can be described theoretically at two different levels: the molecular level and the continuum level. The two different levels are related to the typical length scale involved. The continuum model can describe most transport phenomena in micromixers with a length scale ranging from micrometers to centimeters [40]. The present micromixers were in this range of length scale.

In this study, the fluid entering the two inlets were water and dyed water. They were assumed to have the same physical properties of water at a temperature of 20 °C; the fluid A at Inlet 1 was the dyed water and the fluid B at Inlet 2 was water. The flow inside the micromixer was assumed to be steady, incompressible, laminar, and Newtonian. Therefore, the fluid flow in the micromixer was simulated solving the incompressible Navier–Stokes and the continuity equations:(1)$$\left(\vec{u}\cdot \nabla \right)\vec{u}=-\frac{1}{\rho }\nabla p+\nu \nabla^{2}\vec{u}$$
(2)$$\nabla \cdot \vec{u}=0$$
where *ρ*, $\vec{u}$, *p*, and ν are the density, the velocity vector, the pressure, and the kinematic viscosity of the fluid, respectively. The density and viscosity were specified as 998 kg/m^3^ and 0.001 kg/(ms), respectively.

The flow field thus obtained was used to simulate the mixing process throughout the micromixer. In this study, the mixing was assumed to be governed by two fluid dynamic mechanisms: molecular diffusion and advection. A scalar advection-diffusion transport equation was used to simulate the mixing process [33,37]:(3)$$\left(\vec{u}\cdot \nabla \right)\phi =D\nabla^{2}\phi$$
where *D* and ϕ are the diffusion coefficient and the dye concentration, respectively. The dyed water concentration is ϕ=1 at Inlet 1, and ϕ=0 at Inlet 2. In general, a concentration gradient is a density gradient which can induce a convective flow. The flow velocity due to concentration gradients can vary from 0.1 to 10 μms^−1^ depending on properties such as the channel dimensions, fluid viscosity, fluid type, gradient magnitude [41]; in this paper, the flow velocity in the micromixer was in the range of 0.001 to 0.14 ms^−1^. Therefore, the convective flow induced by any concentration gradient was neglected in this paper.

We used the commercial software ANSYS^®^ Fluent to solve the governing Equations (1)–(3), and it is based on the finite volume method. In general, a certain amount of numerical diffusion is introduced in discretizing the convective terms, but it can be limited by using a high-order discretization scheme. The non-linear convective terms in Equations (1) and (3) were approximated using the QUICK scheme (quadratic upstream interpolation for convective kinematics), and its theoretical accuracy is third order. The velocity was assumed uniform at the two inlets, while it was assumed fully developed at the outlet. Along all the other walls, the no-slip boundary was applied.

In order to obtain a fully converged solution, every computation was continued until oscillation of the residual of all equations was negligibly small. Some researchers used an adaptive mesh technique to reduce the oscillation [42]. In this study, the residuals of continuity, momentum, and concentration equations oscillated in the range smaller than 10^−11^, 10^−14^, and 10^−8^, respectively; they were sufficiently small to obtain reliable numerical solutions.

Accurate numerical simulation of the mixing in micromixers is still a challenging problem, especially for high Peclet numbers. Some studies do not deal with computational issues related to the mesh dependence of numerical solution. In this paper, a detailed study was conducted in the section of validation of numerical solution. According to Okuducu et al. [43], the accuracy of numerical solutions is also dependent on the type of cells; structured hexahedral cells show the most reliable numerical solution, in comparison with tetrahedral and prism cells. In this paper, all cells for every micromixer layout were structured and hexahedral.

The mixing performance of present micro-mixer was evaluated using the degree of mixing (*DOM*) and mixing energy cost (*MEC*). We calculated the *DOM* in the following form:(4)$$DOM=1-\frac{1}{\xi }\sqrt{\sum_{i=1}^{n}\frac{{\left(\phi_{i}-\xi \right)}^{2}}{n}}$$
where *φ_ι_* and *n* are the mass fraction of fluid A in the *i*th cell and total number of cells, respectively, and *ξ* is 0.5 when the two fluids are completely mixed. The *MEC* is calculated in the following form, and measures the effectiveness of a micro-mixer [44,45]:(5)MEC=∆pρumean2DOMX100
where $u_{mean}$ is the average velocity at the outlet, and ∆p is the pressure difference between the inlet and the outlet.

## 4. Validation of Numerical Study

For high Sc number simulations, numerical diffusion is known to deteriorate the accuracy of simulated results, in general [46,47,48,49]. To minimize the numerical diffusion problem, several approaches can be chosen. These include a particle-based simulation such as the Monte Carlo method [46] or a grid-based method with a small cell Peclet number. Here, the cell Peclet number is $Pe=\frac{U_{cell}l_{cell}}{D}$ where $U_{cell}$ and $l_{cell}$ are the local flow velocity and cell size, respectively. However, these approaches are computationally too expensive to adopt in a study like this paper. Most numerical studies prefer a practical approach to obtain numerical solutions with a reasonable degree of accuracy. To achieve that, a detailed study of grid independence including the grid convergence index (GCI) test is usually adopted [18,29,31]. In this paper, a similar procedure was followed.

To quantitatively validate the present numerical approach, we simulated a passive micro-mixer experimented by Tsai et al. [25]. Figure 3 shows a schematic diagram of the micro-mixer. The two inlets had a rectangular cross section of width 45 μm by depth 130 μm. The micromixer had four baffles of width 45 μm by height 97.5 μm. The fluid was assumed to have the properties of density 997 kg/m^3^, viscosity 0.00097 kg/(ms), and diffusion coefficient 3.6 × 10^−10^ m^2^/s, as reported by Tsai et al. [25]. The numerical simulation was carried out for three different Reynolds numbers of *Re =* 1, 9, and 81 and the results were compared with the corresponding experimental data.

Hexahedral cells were used to mesh the computational domain sketched in Figure 3; they are all structured. Before detailed simulations, a preliminary study was carried out to check the grid independence of numerical solutions for the Reynolds number of 9. Figure 4a shows the dependence of the numerical solution for *Re* = 1, and about 1 million of the computational cells were enough to obtain a numerical solution with a reasonable accuracy. Here, *DOM_T_* stands for the degree of mixing defined by Tsai et al. [25] in the following way:(6)$$DOM_{T}=1-\frac{\sigma_{D}}{\sigma_{D,o}}$$
and
(7)$$\sigma =\sqrt{\frac{1}{n}\overset{n}{\underset{i=1}{\sum }}{\left(\phi_{i}-\phi_{ave}\right)}^{2}}$$
where $\sigma_{D}$ is the standard deviation of ϕ on a cross section normal to the flow, $\sigma_{D,o}$ is the standard deviation at the inlet, and $\phi_{ave}\;$ is the average value of ϕ at a sampled cross section.

Figure 4b compares the simulation results with the corresponding experimental data by Tsai et al. [25] for Reynolds numbers from 1 to 81. The numerical solution and experimental data showed similar behavior as the Reynolds number increased, and the discrepancy was acceptable. The discrepancy between the experimental data and numerical solution was less than 4%, and became smaller as the Reynolds number decreased, and as the Peclet number decreased. The discrepancy is attributed to several factors such as the numerical diffusion, experimental uncertainty, etc.

Prior to carrying out the final simulations, an additional set of preliminary simulations was carried out to determine an appropriate mesh size for the simulation of the present micro-mixer. We used the present micro-mixer of Figure 2d for this study, and simulated for the Reynolds number of *Re* = 3. The size of every hexahedral cell was varied from 2.5 μm to 5 μm. Figure 5 shows the dependence of the calculated *DOM* on the edge size. The deviation of 3.5 μm solution from that of 4 μm was 0.4%. Therefore, 3.5 μm was small enough to obtain grid independent solutions.

Using the preliminary simulation results, the uncertainty of grid convergence was evaluated using the grid convergence index (*GCI*) [50,51]. According to the Richardson extrapolation methodology, the *GCI* is defined as follows:(8)$$GCI=F_{s}\frac{\left|\varepsilon \right|}{r^{p}-1}$$
(9)$$\varepsilon =\frac{f_{coarse}-f_{fine}}{f_{fine}}$$
where *F_s_*, *r*, and *p* are the safety factor of the method, grid refinement ratio, and the order of accuracy of the numerical method, respectively. *f_coarse_* and *f_fine_* are the numerical results obtained with a coarse grid and fine grid, respectively. We specified *F_s_* as 1.25 according to the suggestion by Roache [50]. For the GCI evaluation, the edge size was varied from 2.5 μm to 5 μm, 2.5 μm, 3.5 μm, 4 μm, and 5 μm. Table 1 summarizes the result of the *GCI* test. The *GCI* based on the *DOM* at the outlet was 1% and 0.57% for 4 μm and 3.5 μm, respectively. Therefore, the edge size of 3.5 μm was small enough to obtain numerical solutions with reasonable accuracy.

## 5. Results

A new passive micro-mixer with mixing units stacked in the cross flow direction was proposed, its mixing performance was simulated for Reynolds numbers ranging from 0.1 to 50. The present micro-mixer consisted of eight mixing units, and each mixing unit contained four baffles; the baffles were arranged in three different ways. The mixing units were stacked in four different ways to study their effect on the mixing performance.

The velocity at the two inlets was uniform in the range from 0.0293 mm/s to 146 mm/s, and the corresponding volume flow rates were from 6.33 to 633 μL/min. The mixing performance was evaluated in terms of the *DOM* at the outlet and the corresponding *MEC*.

Figure 6 shows the effects of mixing unit stacking in the cross flow direction on the mixing performance in terms of the *DOM* and the corresponding *MEC*. All of the cross flow stacking layouts resulted in a noticeable variation from the baseline layout, and the variation was prominent in the Reynolds number range from 1 to 30. For the Reynolds numbers less than about one, the two step stacking showed the best performance in terms of *DOM*. For example, the *DOM* of the two step stacking at *Re* = 1 showed a 9% increase from that of the baseline layout. On the contrary, the eight step stacking showed the best *DOM* for the Reynolds numbers larger than about two. For example, the *DOM* of the eight step stacking at *Re* = 20 was enhanced 14% from that of the baseline layout. However, the eight step stacking showed the least effective performance for the Reynolds numbers less than about one, as can be seen in the *MEC* distribution; it required the largest pressure load.

We examined how the mixing unit stacking affected the mixing performance of a passive micro-mixer. Figure 7 shows the distribution of mass concentration of the fluid A at the plane of channel half width *z* = 60 μm for *Re* = 1. The concentration contours showed that a stream of high concentration of fluid A (reddish yellow line in the figure) formed along the center of the flow passage. Its appearance from the second mixing unit suggests that the mixing in the first mixing unit was actively achieved by the baffles. One interesting thing to note is that the stream of high concentration impinged on the baffle in the mixing units, especially in the two step stacking layout. Considering that the convective mixing effects for *Re* = 1 were quite limited, the flow characteristics of baffle impingement is an interesting phenomenon in the diffusion dominant flow regime. The baffle impingement seemed most intensive for the case of two step stacking. To quantitatively check this explanation, the *DOM* increment in each mixing unit was compared for the baseline and two step stacking layouts. Figure 8 shows the *DOM* increment in each mixing unit for *Re* = 1. Here, the *DOM* increment means the difference of *DOM* between at the exit and entry of each mixing unit. For example, the flow entry and exit in the second mixing unit of the two step layout are indicated as the arrows pointing upward and downward (see Figure 2c), respectively. The entry and exit *DOMs* were calculated at the corresponding cross section normal to the flow. The increment of the two step stacking in the third and fourth mixing units was noticeably larger than that of the baseline layout. For example, the *DOM* increment in the third mixing unit of two step stacking was about 64% larger than the baseline layout.

Figure 9 shows the distribution of mass concentration of the fluid A at the plane of channel half width *z* = 60 μm for *Re* = 20. The concentration contours showed a very chaotic distribution in the first and second mixing units. However, the stream of fluid A (red stream in the figure) and fluid B (blue stream in the figure) started to segregate from each other in the third mixing unit for the baseline layout. On the other hand, the eight step stacking layout showed a quite different flow pattern. The stream of fluid B impinged on the baffle in the third and fourth mixing units while the stream of fluid A impinged on the baffle in the second, fifth, and sixth mixing units. This suggests that the stream of high concentration changes their position as it flows down along the mixing unit. Figure 10 plots the distribution of mass concentration of the fluid A at the cross sections before and after each mixing unit for the eight step stacking layout, with contours in the *zx* plane (in the cross flow direction). The streams of fluid A and B showed a swirl motion as they passed through the mixing units. Since the swirl motion was in the y-direction (*zx* plane), the stacking of the mixing units in the cross flow direction contributed to the mixing enhancement discussed in Figure 6. This suggests that the mixing enhancement in the convection dominant flow regime can be easily obtained by a simple stacking method of the mixing units in the cross flow direction.

Figure 11 shows the velocity vector field at the junction of third and fourth mixing units (at section (d) in Figure 10) for *Re* = 20, 5, and 2. The velocity vector field in Figure 11a shows two counter rotating vortices. A larger and stronger vortex flow was observed on the lower right corner, and the other vortex was on the upper left corner. Comparing this with Figure 10d, the vortex flow pattern caused a stream of high concentration of fluid A (a red stream on the lower right corner) swirling in the clockwise direction along the mixing unit walls. However, this swirl motion became weak as the Reynolds number decreased, as can be seen in Figure 11b,c. A weak vortex seemed to form near the lower right corner for *Re* = 5, while no explicit pattern of vortex was observed for *Re* = 2. This suggests that the mixing unit stacking in the cross flow direction generated a swirl motion inside the mixing unit, even if the swirl motion was insignificant for Reynolds numbers less than about five. The swirl motion inside the mixing unit is the second flow characteristic attributed to the mixing enhancement described in the Figure 8.

Figure 12 compares the streamlines starting from the two inlets and concentration contours at a plane within each mixing unit for the baseline and eight step stacking layouts for *Re* = 20. The streamline plots were obtained at the cross section of half width *z* = 60 µm. On the right upper side, an enlarged view of the streamlines passing through from the third to fifth mixing units are shown. The concentration contours were obtained at the plane of the first baffle within each mixing unit along the micromixer. The streamlines from the inlet 1 (red) and streamlines from the inlet 2 (blue) changed their relative position more frequently for the eight step stacking layout as they passed down the mixing units; this flow pattern was more clearly seen in the enlarged view shown on the right upper side. The rapid change of their relative position led to faster mixing at the cross section, as can be seen in the centration contours within the mixing units. This flow pattern is also explained by the velocity vector field shown on the right lower side. The eight step stacking layout showed a vortex flow on the right lower corner, and the vector field was wavier than the baseline layout. This vortex flow in the x-direction is another flow characteristic attributed to the stacking of mixing unit in the cross flow direction.

Figure 13 compares the simulated mixing performance in terms of the *DOM* and required pressure load with those from other passive micromixers. In general, the mixing performance of micromixers increased with the diffusion coefficient. The value of diffusion coefficient depends on the fluid used in actual applications. To neglect the effects of the diffusion coefficient on the mixing performance, several passive micromixers based on the same size of the diffusion coefficient were selected for comparison; the diffusion coefficient was 1.2 × 10^−9^ m^2^/s. Tripathi et al. [23] numerically investigated the mixing performance of spiral micromixers and observed transverse flow due to the centrifugal effect. A pair of counter rotating vortices was found to enhance the mixing performance in the spiral micromixer. Raza et al. [29] proposed a SAR mixing unit combined with baffles in a curved channel and simulated its mixing performance. They showed that the mixing performance was better than their earlier version of SAR micro-mixer, and attributed the mixing enhancement to the combined effects of collision, Dean vortex, and secondary flows. Makhsuda et al. [52] proposed an optimal combination of the cross channel split and recombine (CC-SAR) and the mixing cell with baffles (MC-B). They showed that the mixing performance of the optimized micromixer was noticeably improved compared with a micromixer based on the CC-SAR alone. Hossain et al. [53] studied a micromixer with unbalanced three split rhombic sub-channels. According to their results, a rhombic angle of 90° provides the best mixing performance. The mixing performance was enhanced by the two pairs of counter rotating vortices in the cross section. The present micromixer showed a meaningful enhancement of the mixing performance for Reynolds numbers less than about 20, as compared with other micromixers. The enhancement was caused by the two flow characteristics: the baffle impingement by a stream of high concentration and a swirl motion inside the mixing unit. The baffle impingement by a stream of high concentration was observed throughout all Reynolds numbers, while the swirl motion inside the mixing unit became significant only for Reynolds numbers larger than about five. Figure 13b shows that the required pressure load of the present micromixer was comparable with those of other micromixers; the mixing enhancement was achieved without an additional pressure load. This confirms that the mixing unit stacking in the cross flow direction is a practical approach to enhance the mixing performance, especially for Reynolds numbers less than about 20.

## 6. Conclusions

A new passive micro-mixer with the mixing units stacked in the cross flow direction was proposed and estimated. It consisted of eight mixing units, and each mixing unit contained four baffles; the baffles were staggered alternatively in the cross flow and transverse directions. The mixing performance of the present micro-mixer was evaluated numerically by examining the *DOM* and *MEC*. The numerical simulation was carried out for Reynolds numbers ranging from 0.5 to 50.

The numerical solutions showed that the mixing unit stacking in the cross flow direction noticeably enhanced the mixing performance for Reynolds numbers less than about 20, as compared with other micromixers. The stacking layout would be different according to the Reynolds number. For Reynolds numbers less than about two, the two step stacking in the cross flow direction showed the best mixing performance. On the other hand, the eight step stacking showed the best mixing performance for Reynolds numbers larger than about two.

The mixing enhancement due to the mixing unit stacking in the cross flow direction was caused by two different flow characteristics. One is the baffle impingement by a stream of high concentration. A stream of high concentration was observed to impinge on the baffle wall throughout all Reynolds numbers. It was observed even in the limited convection flow regime such as *Re* = 0.5. The other one is the swirl motion inside the mixing unit in the cross flow direction, and it was significant for Reynolds numbers larger than about five.

Since both halves of the present micro-mixer in the transverse direction are planar, they can be simply stacked to construct the present micromixer. The present micromixer performed better than other passive micro-mixers for low Reynolds numbers less than about 20. Therefore, it is expected to be useful as a necessary part of lab-on-a-chip devices and micro-total analysis systems.

## Figures and Tables

**Figure 1 micromachines-12-01530-f001:**
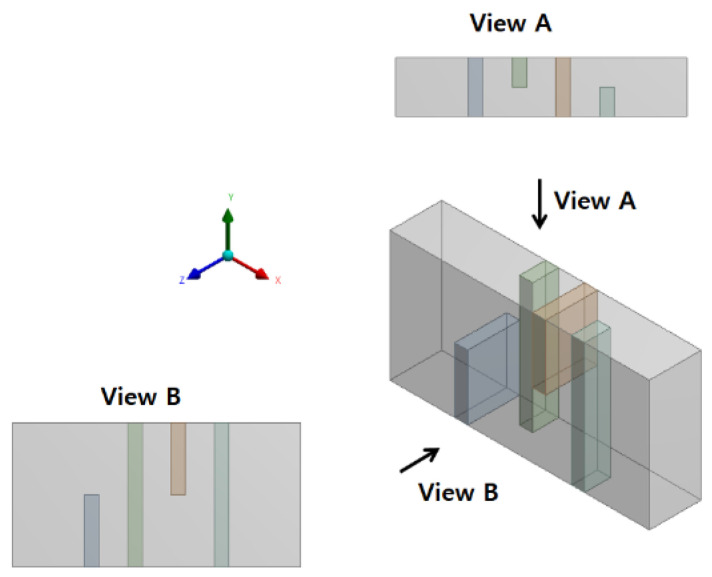
Schematic diagram of the four-baffle arrangement within a mixing unit.

**Figure 2 micromachines-12-01530-f002:**
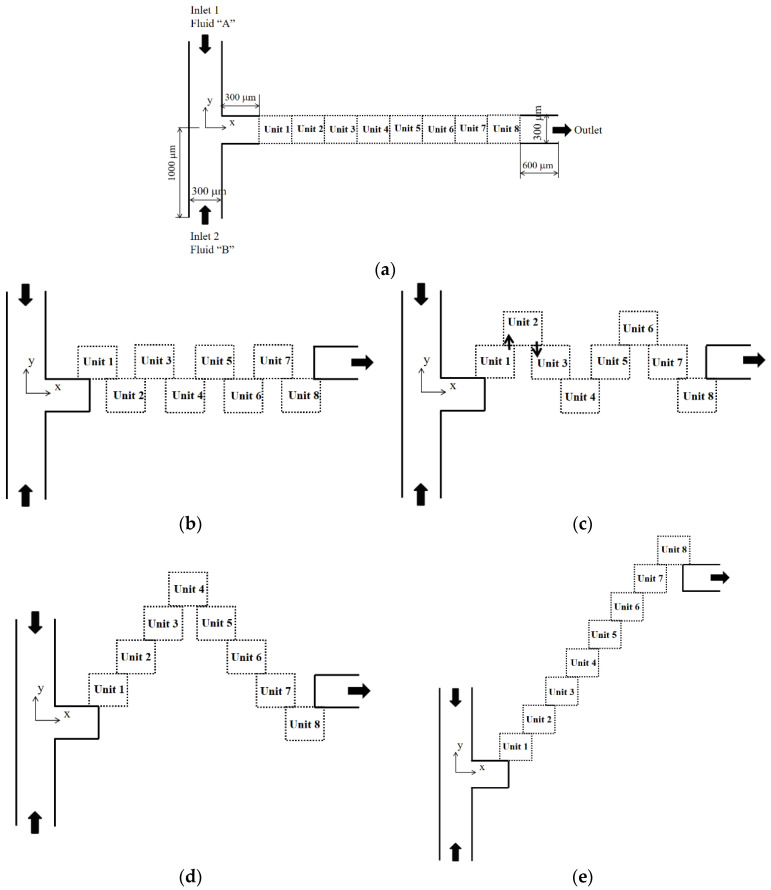
Present passive micro-mixers with mixing units stacked in the cross flow direction. (**a**) Baseline layout. (**b**) One step stacking. (**c**) Two step stacking. (**d**) Four step stacking. (**e**) Eight step stacking.

**Figure 3 micromachines-12-01530-f003:**
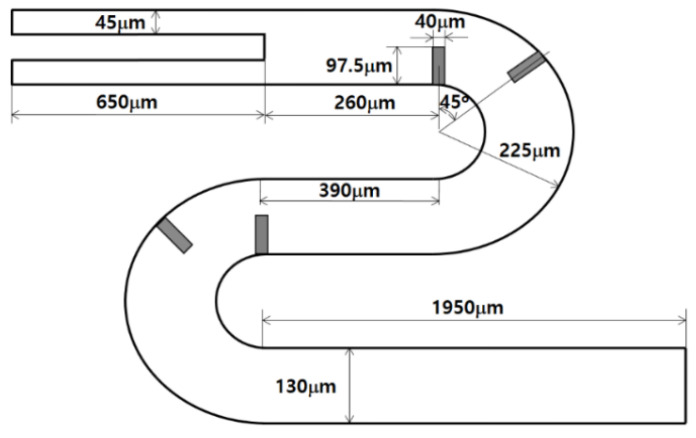
Diagram of the micromixer experimented by Tsai et al. [25].

**Figure 4 micromachines-12-01530-f004:**
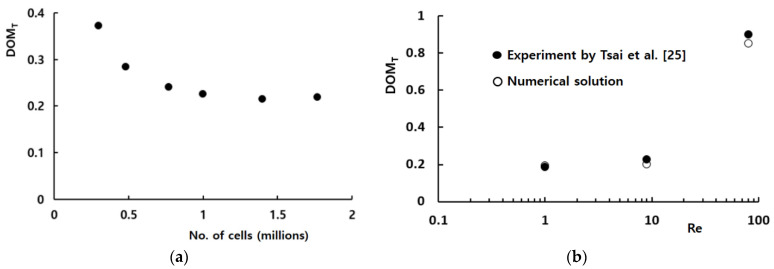
Validation of numerical solution. (**a**) Grid dependence of the numerical solution. (**b**) *DOM* vs. *Re*.

**Figure 5 micromachines-12-01530-f005:**
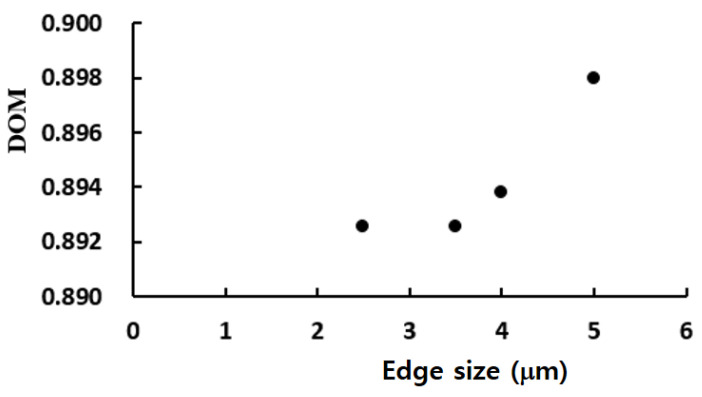
Dependence of the numerical solution on the edge size.

**Figure 6 micromachines-12-01530-f006:**
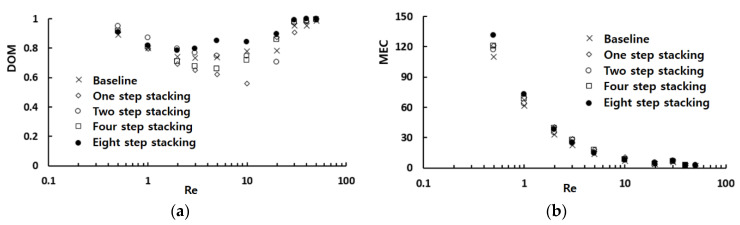
Effects of mixing unit stacking on the mixing performance. (**a**) *DOM* vs. *Re*. (**b**) *MEC* vs. *Re*.

**Figure 7 micromachines-12-01530-f007:**
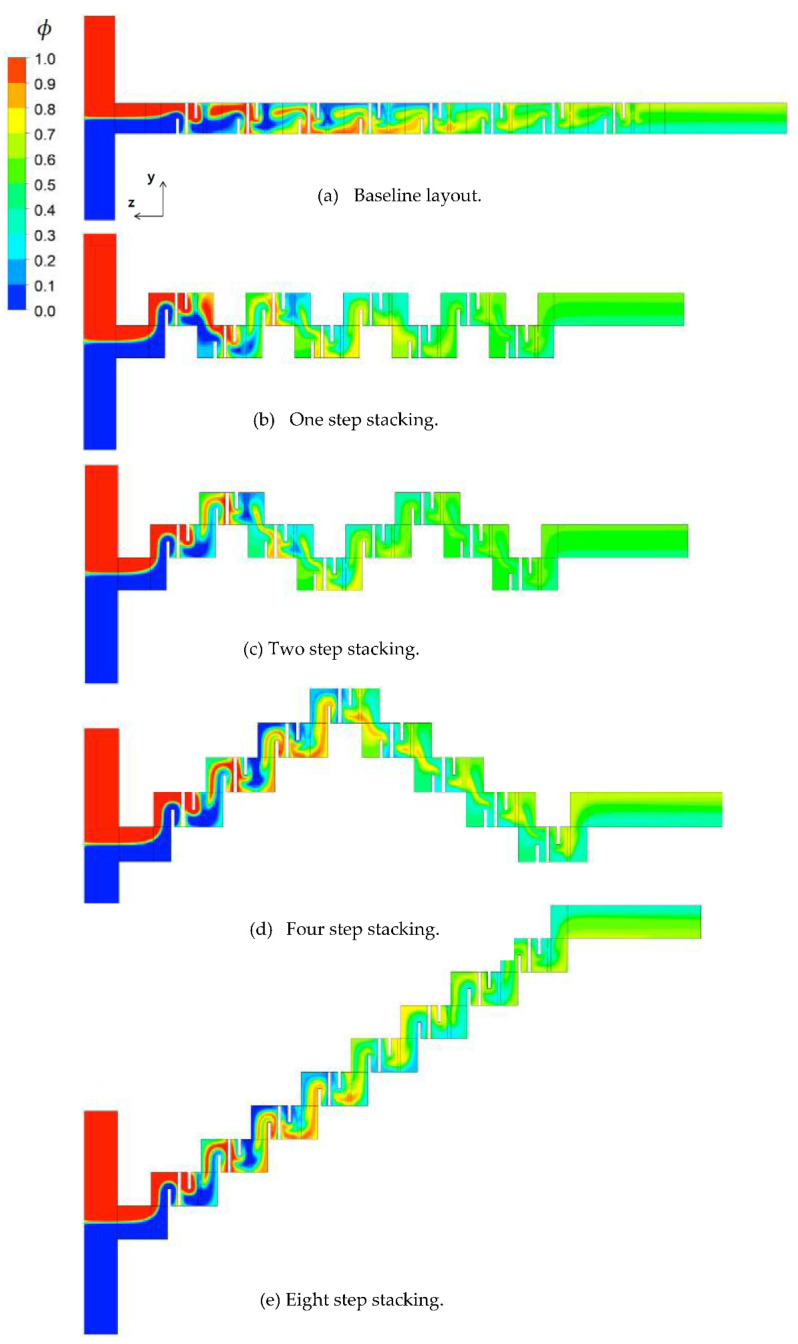
Concentration distribution for *Re* = 1 at the plane of half width *z* = 60 μm.

**Figure 8 micromachines-12-01530-f008:**
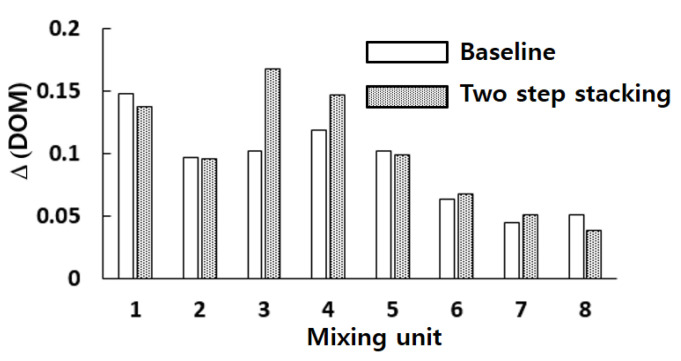
Comparison of *DOM* increment in each mixing unit for *Re* = 1.

**Figure 9 micromachines-12-01530-f009:**
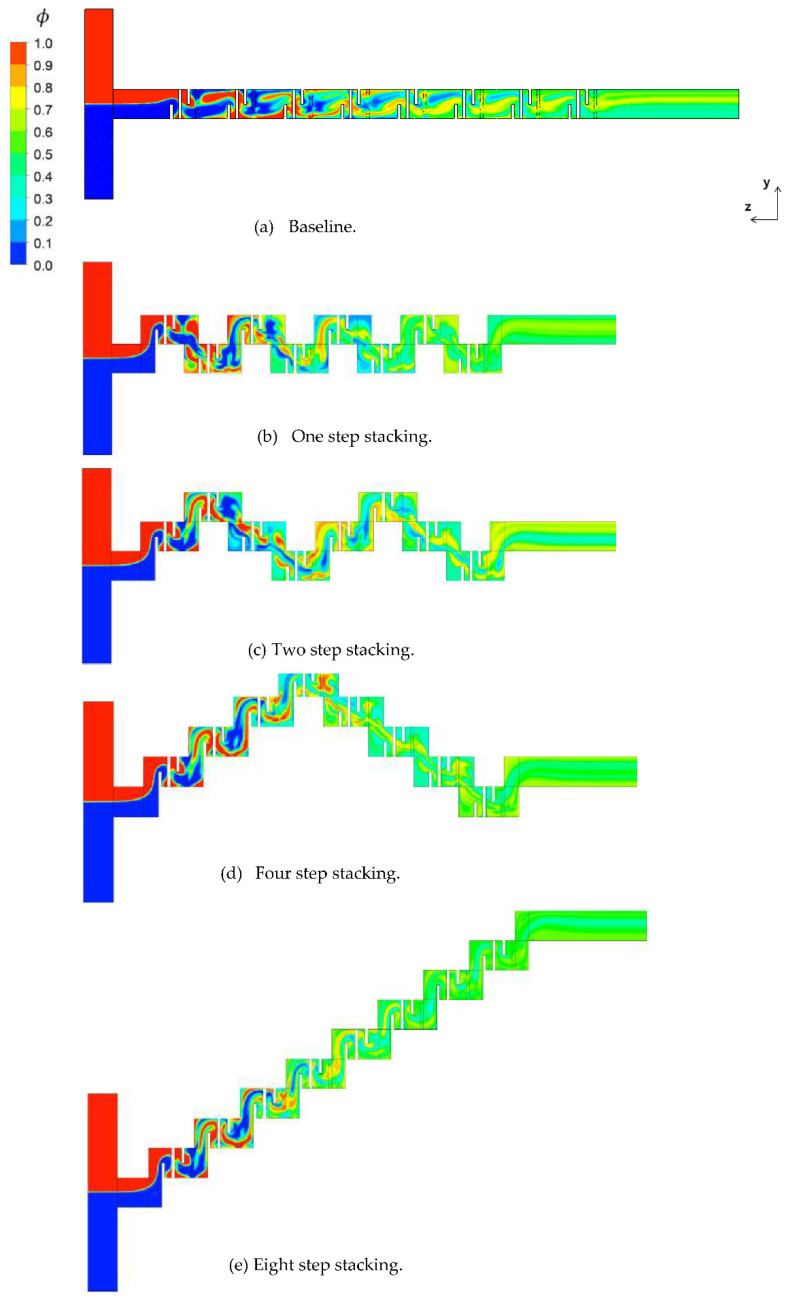
Concentration distribution for *Re* = 20 at the plane of half width *z* = 60 μm.

**Figure 10 micromachines-12-01530-f010:**
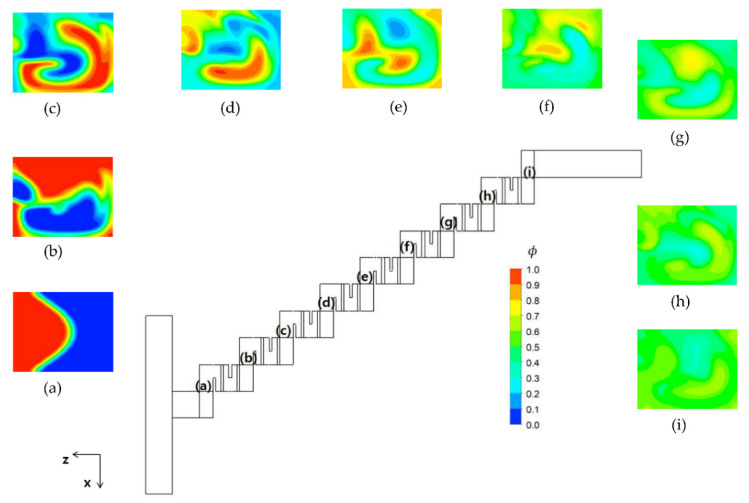
Concentration distribution at the cross section before and after the mixing unit for *Re* = 20.

**Figure 11 micromachines-12-01530-f011:**
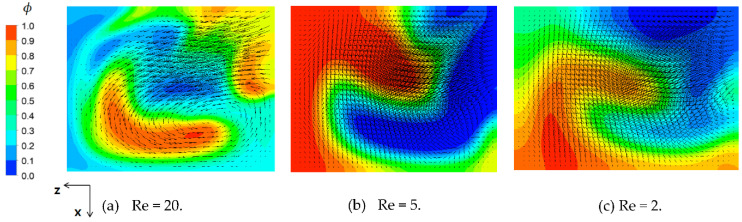
Velocity vector field at the junction of the third and fourth mixing units.

**Figure 12 micromachines-12-01530-f012:**
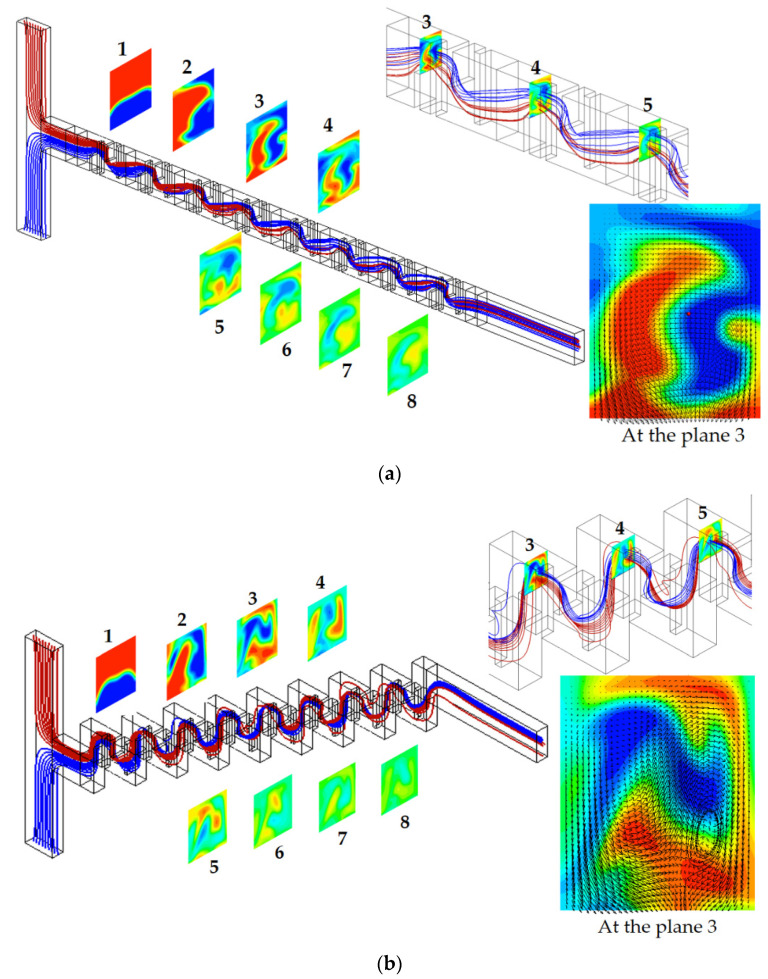
Streamlines starting from the two inlets and concentration contours at the plane of the first baffle within each mixing unit for *Re* = 20. (**a**) Baseline. (**b**) Eight step stacking.

**Figure 13 micromachines-12-01530-f013:**
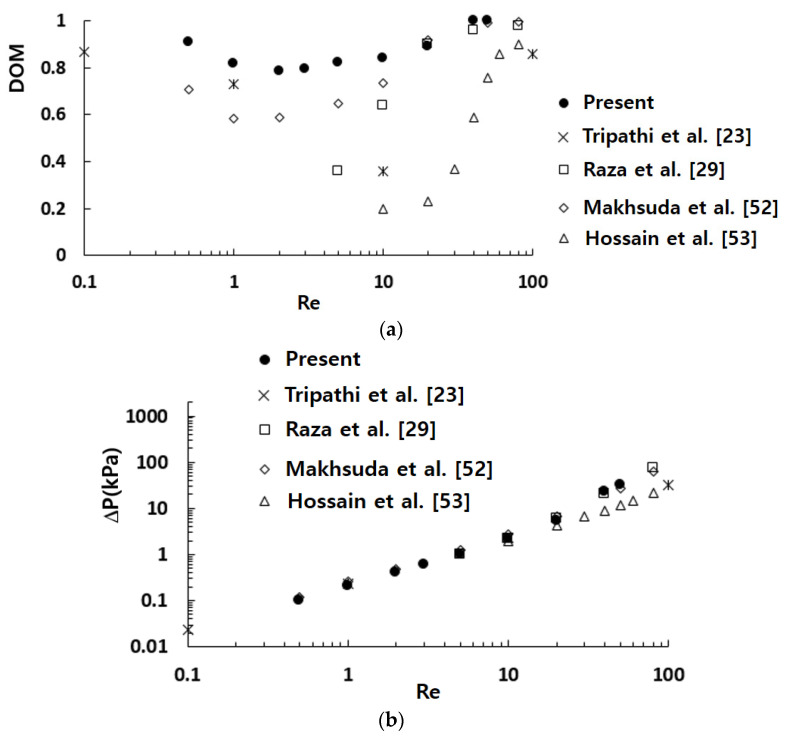
Comparison of the simulated mixing performance with other micromixers. (**a**) *DOM* vs. *Re*. (**b**) Pressure load vs. *Re*.

**Table 1 micromachines-12-01530-t001:** Results of the GCI test.

Edge Size (μm)	*DOM*	*ε*	r	*GCI*
2.5	0.89253	0.00001	1.4	0.00001
3.5	0.89254	0.0014	1.14286	0.00572
4	0.89379	0.00466	1.25	0.01036
5	0.89796

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
