# Peer review of "Mixing Performance of a Passive Micro-Mixer with Mixing Units Stacked in Cross Flow Direction"

_micromachines, 2021, doi:10.3390/mi12121530_

Round 1

Reviewer 1 Report

The article can be accepted for publications, if the following issues be carefully addressed.

  1. It would be interesting to show the different characteristic scales of the studied flows, such as the scales where the mixing takes place or the diffusion times.
  2. The governing equations are not clear. More details must be provided to have enough elements to produce the simulations. Indeed, it is not obvious how the two fluids interact with each other. One guesses that only one advection diffusion is considered since only one diffusivity is given from which we do not know the source. However, it is not clear whether the conservation of mass and momentum equations are solved for each species. It is important to state all the considered equations since they govern how the fluid mix.
  3. Since the concentration is different on the upper and lower half of the channel, their initially remains a concentration gradient that may drive a flow in the transverse direction. The authors can explain the flow regime where the two-way coupling between the concentration and the velocity is neglected in this study.
  4. Figures are not impressive. It is suggested to improve the quality of figures.
  5. Caption of figure 4 should be changed.
  6. The authors should explain the novelty of their work i.e., how their work is different from other.
  7. It is suggested to elaborate the numerical section.
  8. Physically justify the considered range of Reynolds number in the manuscript.
  9. I would also suggest that the authors should include a very useful work on mixing in introduction for the best understanding of readers through different studies, for instance
  • https://doi.org/10.1002/elps.202000225
  • https://doi.org/10.1016/j.jnnfm.2017.06.010
  • https://doi.org/10.1016/j.cep.2018.12.011
  • https://doi.org/10.1016/j.cep.2021.108585
  • https://doi.org/10.1016/j.cherd.2018.06.027
  • https://doi.org/10.1016/j.cep.2021.108609
  • https://doi.org/10.1016/j.cep.2018.03.026
  • https://doi.org/10.1016/j.cep.2021.108335
  • https://doi.org/10.1016/j.apm.2014.12.050
  • https://doi.org/10.1177/0954408919826748
  • https://doi.org/10.1007/s10404-018-2128-3
  1. Several grammatical and typographical errors are there in the paper.

Author Response

Thanks for reviewing the paper.

1) It would be interesting to show the different characteristic scales of the studied flows, such as the scales where the mixing takes place or the diffusion times.

2) The governing equations are not clear. More details must be provided to have enough elements to produce the simulations. Indeed, it is not obvious how the two fluids interact with each other. One guesses that only one advection diffusion is considered since only one diffusivity is given from which we do not know the source. However, it is not clear whether the conservation of mass and momentum equations are solved for each species. It is important to state all the considered equations since they govern how the fluid mix.

3) Since the concentration is different on the upper and lower half of the channel, their initially remains a concentration gradient that may drive a flow in the transverse direction. The authors can explain the flow regime where the two-way coupling between the concentration and the velocity is neglected in this study.

Answer> As you suggested, the manuscript was revised (lines 154~176):

Transport phenomena in micromixers can be described theoretically at two different levels: molecular level and continuum level. The two different levels of description correspond to the typical length scale involved. Continuum model can describe most transport phenomena in micromixers with a length scale ranging from micrometers to centimeters [38]. Present micromixers are applicable in this range of length scale.

The fluid entering the two inlets are water and dyed water, and they are assumed to have the same physical properties of water at a temperature of 20oC; the fluid A at Inlet 1 is the dyed water and the fluid B at Inlet 2 is water. Therefore, the fluid flow in the micromixer was simulated solving the incompressible Navier–Stokes and the continuity equations;

  (1)                                             

   (2)                                                 

where r, , p, and ν are the density, the velocity vector, the pressure, and the kinematic viscosity of the fluid, respectively. The density and viscosity are specified as 998 kg/m3 and 0.001 kg/(ms), respectively.

The flow field thus obtained are used to simulate the mixing process throughout the micromixer. In this study, the mixing was assumed to be governed by two fluid dynamic mechanisms: molecular diffusion and advection. A scalar advection-diffusion transport equation was used to simulate the mixing process [33, 37];

   (3)

where D and   are the diffusion coefficient and  the dye concentration, respectively. The dyed water concentration is f  = 1 at Inlet 1, and  f  = 0  at Inlet 2. In general, a concentration gradient is a density gradient which can induce a convective flow. The flow velocity due to concentration gradients can vary from 0.1 to 10 μms-1, depending on properties such as the channel dimensions, fluid viscosity, fluid type, gradient magnitude [41]; in this paper, the flow velocity is in the range of 0.001 to 0.14 ms-1. Therefore, the convective flow induced by any concentration gradient is neglected in this paper.

4) Figures are not impressive. It is suggested to improve the quality of figures.

Answer> As you suggested, some figures were revised (Figures 7, 9, 10, and 11):

5) Caption of figure 4 should be changed.

Answer> The caption is corrected:

Grid dependence of numerical => Grid dependence of numerical solution.

6) The authors should explain the novelty of their work i.e., how their work is different from other.

Answer> As you suggested, the manuscript was revised (lines 68-79):

Recently, several complicated three-dimensional (3D) micro-mixers showed an improved mixing performance, compared with those of two-dimensional (2D) micro-mixers of similar size [30~33]. However, 2D planar micromixers have an advantage of simplicity in fabrication compared to that of complex 3D micromixers. Various design techniques were attempted to promote 3D effects: baffles, channel wall twisting, spiral channel and split and recombine. For example, Kang [13] showed that a cyclic configuration of baffles generates a rotational flow in the cross section of a micromixer. The same idea was adopted in the present micromixer to promote 3D effects, and four baffles are attached to the four side walls of the micromixer in a cyclic order. The overall layout of a passive micromixer is also an important design concept [33]. For example, Tripathi [23] studied three different layout of a passive micromixer based on spiral microchannel, and showed that the mixing performance is quite dependent on the micromixer layout. The eight mixing units are stacked in the cross flow direction; this kind of stacking has not been studied yet. The stacking was designed as four different layouts: one step, two step, four step, and eight step stacking. The combined effects of a cyclic baffle arrangement and mixing unit stacking in the cross flow direction is the main design concept of present micromixer.

7) It is suggested to elaborate the numerical section.

Answer> As you suggested, the manuscript was revised (lines 176-193):

We used the commercial software ANSYS® Fluent to solve the governing Equations (1)-(3), and it is based on the finite volume method. In general, a certain amount of numerical diffusion is introduced in discretizing the convective terms, but it can be limited by using a high-order discretization scheme. The non-linear convective terms in Equations (1) and (3) were approximated using the QUICK scheme (quadratic upstream interpolation for convective kinematics), and its theoretical accuracy is third order. The velocity was assumed uniform at the two inlets, while it was assumed fully developed at the outlet. Along all the other walls, the no-slip boundary was applied.

In order to obtain a fully converged solution, every computations were continued until oscillation of the residual of all equations is negligibly small. Some researchers used an adaptive mesh technique to reduce the oscillation [42]. In this study, the residuals of continuity, momentum, and concentration equations oscillated in the range smaller than 10-11, 10-14, and 10-8, respectively; they are sufficiently small to obtain reliable numerical solutions.

Accurate numerical simulation of the mixing in micromixers is still a challenging problem especially for high Peclet numbers. Some studies do not deal with computational issues related to the mesh independence of numerical solution. According to Okuducu et al. [43], the accuracy of numerical solutions is also dependent on the type of cells; structured hexahedral cells show the most reliable numerical solution, in comparison with tetrahedral and prism cells. In this paper, all cells for every micromixer layouts are structured and hexahedral.

8) Physically justify the considered range of Reynolds number in the manuscript.

Answer> As you suggested, the manuscript was revised (lines 80-87):

Most of micromixers used in the biological and chemical applications operate usually in the range of millisecond mixing time, and the corresponding Reynolds number is less than about 100 [34~36]; the corresponding volume flow rate is an order of mL/h. In this range, the mixing mechanism can be divided into three regimes: molecular dominance, transition, and advection dominance [10, 18, and 33]. The effects of micromixer layout design as well as a passive device such as baffles are known to be significant in all of the three regimes. Accordingly, present numerical study was carried out for the Reynolds numbers from 0.5 to 50, covering all of the three regimes. The corresponding volume flow rate ranges from 6.33 μL/min to 633 μL/min.

9) I would also suggest that the authors should include a very useful work on mixing in introduction for the best understanding of readers through different studies.

Answer> As you suggested, the manuscript was revised:

  1. Mondal, B.; Metha, S.K.; Pati, S.; Patowari, P.K. Numerical analysis of electroosmotic mixing in a heterogeneous charged micromixer with obstacles, Eng. Proc.-Process intensification, 2021, 168, 108585.
  2. Gunti, K.; Bhattacharya, A.; Chakraborty, S. Analysis of micromixing of non-Newtonian fluids driven by altering current electrothermal flow, Non-Newtonian fluid mechanics, 2017, 247, 123-131.
  3. Mondal, B.; Mehta, S.K.; Patowari, P.K.; Pati, S. Numerical study of mixing in wavy micromixers: comparison between raccoon and serpentine mixer, Eng. Proc.-Process intensification, 2019, 136, 44-61.
  4. Tripathi, E.; Patowari, P.K.; Pati, S. Numerical investigation of mixing performance in spiral micromixers based on Dean flows and chaotic advection, Eng. Proc.-Process intensification, 2021, 169, 108609.
  5. Borgohain, P; Arumughan, J.; Dalal, A.; Natarajan, G. Design and performance of a three-dimensional micromixer with curved ribs, Chem. Eng. Res. Design, 2018, 136, 761-775.
  6. Hessel, V.; LÖwe, H.; SchÖnfeld, F. Micromixers-a review on passive and active mixing principles, Eng. Sci., 2006, 60, 2479-2501.
  7. Liao, Y.; Mechulam, Y.; Lassale-Kaiser, B. A millisecond passive micromixer with low flow rate, low sample consumption and easy fabrication, Science Reports, 2021, 11-20119.
  8. Cook, K. J.; Fan, Y.F.; Hassan, I. Mixing evaluation of a passive scaled-up serpentine micromixer with slanted grooves, ASME, J. Fluids Eng., 2013, 135(8), 081102.

Reviewer 2 Report

The authors present an interesting numerical study of different designs of a passive micromixer consisting of 8 unit cells with baffles. Each design reflects a different stacking of the unit cells. The computations are performed for Re from 0.5 to 50 (flow rates from 6.33 μL/min to 633 μL/min). The mixing performance is evaluated with the degree of mixing (DOM) and mixing energy cost (MEC). The results of the comparative study demonstrate that depending on Re, a different stacking is optimum. Although the study is systematic, there are still clarifications and amendments that need to be addressed by the authors, according to the following comments:

1) The mixing unit with the 4 baffles is the central element of the authors’ proposal. Why do they use the specific type of (4) baffles in the mixing unit? What is the motivation behind this specific choice?

2) The diffusion coefficient of biomolecules is less than 10-9 m2/s. In previous studies on mixing in the context of lab-on-chip systems, values of diffusion coefficient equal to 10-10 m2/s (1) and 10-11 m2/s (2) were used. The authors should comment on using this value for the diffusion coefficient. A lower value would induce a more valuable comparison among the different designs at realistic conditions.

3) Line 141, please substitute μ with ν.

4) The mixing efficiency, equivalent to the term degree of mixing used by the authors, is calculated at a cross-section normal to the flow direction see Eq. 6 [paper of Tsai et al. (ref 20 of the submitted study) as well as the studies in refs. (1-4)]. Nevertheless, the authors calculate the mixing efficiency at a unit cell; I am not sure whether this is a representative measure of the mixing efficiency.

5) It is very good, although not common in similar studies, that the authors perform a mesh independence study. Without a mesh independence study, the numerical results are of no usefulness. This should be pointed in the text.

6) In mixing problems in microfluidics, i.e., at laminar flow conditions and under low diffusion coefficient of the solute, there are difficulties in the numerical solution at coarse meshes, especially in cases of high Pe. The result of these difficulties is oscillations near the steep changes of the solute concentration. To deal with this issue, the mesh must be adequately refined, and stabilization techniques are utilized. Adaptive mesh techniques are proposed (1) to reduce the degrees of freedom and the computational cost. Did the authors observe these oscillations? Did they use an adaptive mesh?

7) Figure 13: Unless the diffusion coefficients are equal, the mixing efficiencies among different studies cannot be compared. Is the diffusion coefficient of the solute equal to all compared studies?

8) The authors conclude that the proposed designs of the micromixers are expected to be useful in a variety of applications. How would the proposed micromixer be fabricated? It is a 3D design (it cannot be fabricated with standard lithographic techniques) with dimensions at the microscale.

References

  1. Kefala I, Papadopoulos V, Karpou G, Kokkoris G, Papadakis G, Tserepi A. A labyrinth split and merge micromixer for bioanalytical applications. Microfluid Nanofluid. 2015;19(5):1047-59.
  2. Hadjigeorgiou AG, Boudouvis AG, Kokkoris G. Thorough computational analysis of the staggered herringbone micromixer reveals transport mechanisms and enables mixing efficiency-based improved design. Chemical Engineering Journal. 2021;414:128775.
  3. Cai G, Xue L, Zhang H, Lin J. A Review on Micromixers. Micromachines. 2017;8(9):274.
  4. Nguyen N. Micromixers. fundamentals, design and fabrication. Norwich, NY: William Andrew; 2008.

Author Response

Thanks for reviewing the paper.

1) The mixing unit with the 4 baffles is the central element of the authors’ proposal. Why do they use the specific type of (4) baffles in the mixing unit? What is the motivation behind this specific choice?

Answer> As you suggested, the manuscript was revised (lines 68~79):

However, 2D planar micromixers have an advantage of simplicity in fabrication compared to that of complex 3D micromixers. Various design techniques were attempted to promote 3D effects: baffles, channel wall twisting, spiral channel and split and recombine. For example, Kang [13] showed that a cyclic configuration of baffles generates a rotational flow in the cross section of a micromixer. The same idea was adopted in the present micromixer to promote 3D effects, and four baffles are attached to the four side walls of the micromixer in a cyclic order. The overall layout of a passive micromixer is also an important design concept [33]. For example, Tripathi [23] studied three different layout of a passive micromixer based on spiral microchannel, and showed that the mixing performance is quite dependent on the micromixer layout. The eight mixing units are stacked in the cross flow direction; this kind of stacking has not been studied yet. The stacking was designed as four different layouts: one step, two step, four step, and eight step stacking. The combined effects of a cyclic baffle arrangement and mixing unit stacking in the cross flow direction is the main design concept of present micromixer.

2) The diffusion coefficient of biomolecules is less than 10-9m2/s. In previous studies on mixing in the context of lab-on-chip systems, values of diffusion coefficient equal to 10-10 m2/s (1) and 10-11 m2/s (2) were used. The authors should comment on using this value for the diffusion coefficient. A lower value would induce a more valuable comparison among the different designs at realistic conditions.

Answer>

As you suggested, the manuscript was revised (lines 485~489):

Figure 13 compares the simulated mixing performance in terms of the DOM and required pressure load with those from other passive micromixers. In general, the mixing performance of micromixers increases with the diffusion coefficient. The value of diffusion coefficient depends on the fluid used in actual applications. To neglect the effects of the diffusion coefficient on the mixing performance, several passive micromixers based on the same size of the diffusion coefficient were selected for comparison; the diffusion coefficient is 1.2X10-9 m2s-1.

3) Line 141, please substitute μ with ν.

Answer> As you pointed out, the manuscript was revised (line 118):

The Reynolds number is defined as , where  indicate the mean velocity at the outlet, the hydraulic diameter of outlet, and the kinematic viscosity of the fluid, respectively.

4) The mixing efficiency, equivalent to the term degree of mixing used by the authors, is calculated at a cross-section normal to the flow direction see Eq. 6 [paper of Tsai et al. (ref 20 of the submitted study) as well as the studies in refs. (1-4)]. Nevertheless, the authors calculate the mixing efficiency at a unit cell; I am not sure whether this is a representative measure of the mixing efficiency.

Answer> The degree of mixing (DOM) was calculated at the cross section normal to the flow direction.

Anyway, the manuscript was revised for better understanding (lines 356~359):

To quantitatively check this explanation, the DOM increment in each mixing unit was compared for the baseline and two step stacking layouts. Figure 8 shows the DOM increment in each mixing unit for Re = 1. Here, the DOM increment means the difference of DOM between at the exit and entry of each mixing cell. For example, the flow entry and exit in the second mixing unit of the two step layout are indicated as the arrows upward and downward (see Figure 2c), respectively. The entry and exit DOMs were calculated at the corresponding cross section normal to the flow.

5) It is very good, although not common in similar studies, that the authors perform a mesh independence study. Without a mesh independence study, the numerical results are of no usefulness. This should be pointed in the text.

6) In mixing problems in microfluidics, i.e., at laminar flow conditions and under low diffusion coefficient of the solute, there are difficulties in the numerical solution at coarse meshes, especially in cases of high Pe. The result of these difficulties is oscillations near the steep changes of the solute concentration. To deal with this issue, the mesh must be adequately refined, and stabilization techniques are utilized. Adaptive mesh techniques are proposed (1) to reduce the degrees of freedom and the computational cost. Did the authors observe these oscillations? Did they use an adaptive mesh?

Answer> As you suggested, the manuscript was revised (lines 183~194):

In order to obtain a fully converged solution, every computations were continued until oscillation of the residual of all equations is negligibly small. Some researchers used an adaptive mesh technique to reduce the oscillation [42]. In this study, the residuals of continuity, momentum, and concentration equations oscillated in the range smaller than 10-11, 10-14, and 10-8, respectively; they are sufficiently small to obtain reliable numerical solutions.

Accurate numerical simulation of the mixing in micromixers is still a challenging problem especially for high Peclet numbers. Some studies do not deal with computational issues related to the mesh dependence of numerical solution. In this paper, a detailed study was conducted in the section of validation of numerical solution. According to Okuducu et al. [43], the accuracy of numerical solutions is also dependent on the type of cells; structured hexahedral cells show the most reliable numerical solution, in comparison with tetrahedral and prism cells. In this paper, all cells for every micromixer layouts are structured and hexahedral.

7) Figure 13: Unless the diffusion coefficients are equal, the mixing efficiencies among different studies cannot be compared. Is the diffusion coefficient of the solute equal to all compared studies?

Answer> As you suggested, the comparison was made for the same value of diffusion coefficient (lines 486~490):

Figure 13 compares the simulated mixing performance in terms of the DOM and required pressure load with those from other passive micromixers. In general, the mixing performance of micromixers increases with the diffusion coefficient. The value of diffusion coefficient depends on the fluid used in actual applications. To neglect the effects of the diffusion coefficient on the mixing performance, several passive micromixers based on the same size of the diffusion coefficient were selected for comparison; the diffusion coefficient is 1.2X10-9 m2s-1.

8) The authors conclude that the proposed designs of the micromixers are expected to be useful in a variety of applications. How would the proposed micromixer be fabricated? It is a 3D design (it cannot be fabricated with standard lithographic techniques) with dimensions at the microscale.

Answer> As you suggested, the manuscript was revised (lines 548~551):

Since both half of the present micro-mixer is planar in the cross flow direction, they can be simply stacked to construct the present micromixer. The present micromixer performs better than other passive micro-mixers for the low Reynolds numbers less than about 20. Therefore, it is expected to be useful as a necessary part of lab-on-a-chip devices and micro-total analysis systems.

Round 2

Reviewer 1 Report

I am satisfied with the revised version of the manuscript and recommend it for publication.